# Music Is Served: How Acoustic Interventions in Hospital Dining Environments Can Improve Patient Mealtime Wellbeing

**DOI:** 10.3390/foods10112590

**Published:** 2021-10-26

**Authors:** Signe Lund Mathiesen, Lena Aadal, Morten Laulund Uldbæk, Peter Astrup, Derek Victor Byrne, Qian Janice Wang

**Affiliations:** 1Department of Food Science, Faculty of Technical Sciences, Aarhus University, 8200 Aarhus N, Denmark; derekv.byrne@food.au.dk (D.V.B.); qianjanice.wang@food.au.dk (Q.J.W.); 2Hammel Neurorehabilitation and Research Center, 8450 Hammel, Denmark; lena.aadal@clin.au.dk; 3Department of Clinical Medicine, Faculty of Health, Aarhus University, 8000 Aarhus C, Denmark; 4LYdBEHAG, 8850 Bjerringbro, Denmark; morten@lydbehag.dk; 5Test and Development Center for Welfaretech, 8800 Viborg, Denmark; peter.astrup@tucv.dk

**Keywords:** music intervention, sound, eating experiences, multisensory, environmental factors, mealtime wellbeing, rehabilitation, ABI patients, interdisciplinary

## Abstract

Eating-related challenges and discomforts arising from moderately acquired brain injuries (ABI)—including physiological and cognitive difficulties—can interfere with patients’ eating experience and impede the recovery process. At the same time, external environmental factors have been proven to be influential in our mealtime experience. This experimental pilot study investigates whether redesigning the sonic environment in hospital dining areas can positively influence ABI patients’ (*n* = 17) nutritional state and mealtime experience. Using a three-phase between-subjects interventional design, we investigate the effects of installing sound proofing materials and playing music during the lunch meals at a specialised ABI hospital unit. Comprising both quantitative and qualitative research approaches and data acquisition methods, this project provides multidisciplinary and holistic insights into the importance of attending to sound in hospital surroundings. Our results demonstrate that improved acoustics and music playback during lunch meals might improve the mealtime atmosphere, the patient well-being, and social interaction, which potentially supports patient food intake and nutritional state. The results are discussed in terms of potential future implications for the healthcare sector.

## 1. Introduction

Addressing disease-related malnutrition is urgent, and nutrition is key to addressing health and well-being and reducing healthcare costs [1,2]. Malnutrition is associated with increased morbidity, prolonged hospital stays, decreased outcome and higher healthcare costs [3,4,5]. In Denmark, the annual cost of in-hospital malnutrition is estimated at DKK 6 billion [6]; yet, research addressing this challenge during the recovery trajectory is largely non-existent. The present pilot study was undertaken to explore novel avenues of optimising the physical hospital dining environment to improve patient nutrition and mealtime experience.

### 1.1. Background

Patients with an acquired brain injury (ABI) are at risk of malnutrition both in the acute and the subacute rehabilitation phase [7,8,9,10,11,12,13,14,15]. Nutritional care is, thus, pivotal, and by tradition, nurses take responsibility for meeting patients’ nutritional needs [16,17,18,19,20]. Recently, an increasing focus on ensuring aesthetics in order to improve the mealtime experience has been documented [21,22,23,24,25,26,27]. Combined with the existing realisation that environmental surroundings contribute to the meal experience to a great extent [28,29], research exploring the environmental and aesthetic elements that support patient food intake has identified, in particular, that social relations, i.e., dining with others [30,31,32,33] as well as an attractive physical environment [34,35], are of crucial importance, and that these factors have been linked to patients’ feelings of safety, comfort, and well-being [25,36,37].

#### 1.1.1. Visual Meal Environment Aesthetics

Increasing the attractiveness of the care environment may include the display of art works, such as paintings, sculptures, or other installations, either as temporary exhibitions or as permanent integrations within the buildings themselves [38]. Scarce evidence indicates that the presence of art in hospital settings may contribute positively to health outcomes among both patients and staff. Specifically, art interventions have been linked to improvements in stress, mood, pain, and sleep [39,40]. 

One study concluded that images of naturalistic nature scenes compared to either abstract or representational paintings, reduced mental health patients’ anxiety and agitation during 3- to 4-day stays in a United States hospital [41]. Another publication argued that psychological responses to the green and blue colours found in landscapes and natural environments elicited higher levels of pleasure and lower levels of arousal of patients in three Scottish hospitals, thereby, explaining the preference for landscape and nature scenes among patients [39]. Effects of specifically implementing visual art works in meal environments on the meal experience or nutritional status of patients has, to the best of our knowledge, not been examined.

#### 1.1.2. Acoustic Meal Environments

Well-documented evidence shows that the acoustical quality of a room greatly impacts our hearing, such as our ability to interpret acoustic communication modes, e.g., speech [42,43,44,45], in addition to how well music is reproduced and the perceived quality of the listening experience [46]. Room acoustics are determined by the characteristics of reflective surfaces, such as floors, walls, ceilings, and windows. 

Hard, reflective surfaces, often found throughout hospital structures [47], will typically result in an increase of the so-called reverberation time and decrease speech intelligibility, which can contribute to an unfavourable acoustic milieu. Reverberation time (RT) refers to the decay process of the accumulation of sound after the source of the sound has ceased (measured in seconds) and is a significant determinant of the acoustical quality of rooms according to their purpose criteria (ISO, 2009), while speech intelligibility measures the effectiveness of communication in an environment. The most widely used parameter for measuring speech intelligibility is the standardised, objective “speech transmission index” (STI), taking values between 0 and 1. The higher the number, the better the intelligibility of speech. [48].

According to Danish regulations, no specific noise exposure level thresholds exist for hospitals or other healthcare facilities, including common areas, such as dining rooms. However, in an effort to provide some guidelines for room acoustics in hospital canteens and cafeterias, the executive order on building regulations recommends to follow the acoustic criteria of the so-called sound classification C when no regulations are specified [49]. According to this class, values of reverberation time should not exceed 0.6 s, whereas no value for speech transmission index in common rooms is specified [50]. 

Further, the building code states that “it must be ensured with due consideration of the use of the building that persons in the building are not disturbed by sound” [49]. Evidently, decisions regarding sound-based interventions must be made on a case-by-case basis, dependent on the specific properties of the spaces. Acoustical room analyses are, thus, typically required to ensure that any sonic or music intervention within existing architectural structures is appropriate according to the intended purpose of the room.

Although detailed investigations into the role of the acoustic hospital mealtime environment have been largely overlooked in scientific literature, studies of the beneficial and detrimental effects of music and noise on various other health outcomes exist in abundance [51,52]. In particular, numerous accounts of problematic noise levels and/or stressful auditory surroundings at hospitals have been published, expounding the harmful consequences to patient well-being during hospitalisation [47,53,54,55,56,57]. These include, among others, poor sleep [58], increased blood pressure [59], impaired wound healing [60], and general discomfort and annoyance [55,61]. It is thus of great interest to pursue creating satisfactory auditory environments for meal-based activities in hospitals that ensure favourable bases for nutritional care and patient wellbeing.

#### 1.1.3. Mealtime Music

Music is known to be an enriching emotional, cognitive, and physical experience when sought or encountered in our surroundings [62]. The mounting evidence of music’s effectiveness has prompted increasing interest from health care professionals to improve physiological, psychological, and cognitive disorders by means of low-cost, yet effective alternatives to conventional medicine, e.g., in the form of music interventions or “music therapy” [56]. A limited number of studies have examined the effects of using music in healthcare mealtime activities [63]. Results from these studies suggest that playing “relaxation music” during mealtimes eases disruptive behaviour [64] and agitation [65,66,67,68,69,70]; reduces patients’ irritability, anxiousness, and depression [71]; and improves food consumption or caloric intake among the patients [70,71,72].

Although empirical evidence from the above-mentioned studies reveals positive outcomes related to food intake and mealtime behaviour, their general validity is somewhat limited by small and narrow study populations with severe cognitive impairments and/or exhibiting adverse behaviour. Furthermore, outcome measures of a mainly observational nature, e.g., agitation rating scales and checklists, staff surveys, patient weight, and pulse rate recordings were used, neglecting the relevance of patients’ experienced mealtime and curtailing the potential relevance to other patient groups. In addition to this, attention to individuals’ music preferences or responses were insufficiently addressed, and while some did report registrations of background noise levels [64,65,66,69,72], this data appeared to be recorded solely for the purpose of determining music playback volume. 

Other non-musical acoustic properties of the architectural structures were otherwise widely disregarded, which presents challenges for replicating results and raises concerns related to the overall increased sum of sound sources when music is added to an environment [56]. Finally, the music selections used primarily reflect the well-documented genre of “relaxation music” within music therapy practice [73,74,75,76], but it remains to be examined whether this music style is appropriate in other settings and for other patient groups.

### 1.2. Objectives

Dining areas of hospitals are patently complex, vary greatly from facility to facility, and comprise numerous elements, all of which affect the patient mealtime experience. Likewise, mealtime practices will differ according to patient groups, resources, etc. This exposes an indisputable need to at once address the nature of the perceived components in isolation while also taking into consideration the holistic multisensory experience embedded in specific environments. The current pilot study presents an exploratory approach to aestheticising a hospital eating environment. Specifically, we propose that careful attention to room acoustics and wall décor in addition to mealtime music improves nutritional care in a hospital rehabilitation context. In a collaborative effort between audio industry professionals, researchers, and clinicians, this pilot projects seeks to:

Identify and resolve issues in the existing acoustic environment of a common dining area of a hospital ward.Explore how improvements to the acoustic eating environment, including music playback, affects patients’ mealtime experience, behaviour, and food intake.Examine various musical genres and their appropriateness for eating situations in hospital settings.

## 2. Materials and Methods

### 2.1. Design

The project followed a novel methodological interventional framework, incorporating acoustical measurements, observational studies of patients during mealtime, and individual semi-structured interviews with patients and staff. The project was designed as a between-subjects design comparing effects between the three phases, in which each intervention phase builds upon the previous (see Figure 1). 

Phase 1 was a baseline period with no interventions, the purpose of which was to provide a reference point both for the acoustic and music interventions (Phases 2 and 3) as well as a control for mealtime behaviour observations and interview data. Sound proofing materials installed in Phase 2 were chosen based on measurements at baseline, whereas the music playback equipment and volume level in Phase 3 was set according to the room acoustics data obtained from measurements in Phase 2. The study lasted approximately three months, with each intervention running for 17–18 days on weekdays during lunch (11:30–12:15). A designated project nurse was appointed to administer all project activities related to the hospital mealtime processes.

#### 2.1.1. Room Acoustics Analysis, Treatment, and Wall Panel Design

Initial conversations with hospital staff revealed certain annoyances related to the acoustic parameters of reverberation time [42,77] and speech intelligibility [48] in the patient dining area. Specifically, the project nurse characterised the room as “ringing” and described a tendency of the room to accumulate “noise”, which, in turn, caused both staff and patients to gradually increase the volume of their speech in order to make themselves heard. She noted that this type of interaction was poorly suited to certain patients, who could respond negatively by withdrawing from conversation, thus, (inadvertently) excluding themselves from the community.

Prior to the baseline phase, acoustic analysis of reverberation time and speech transmission index in the empty dining room was performed. These measurements revealed an average speech transmission index of 0.604, while the average reverberation time was 0.97 s, exceeding the recommended limit values for sound Class C [50]. Based on these measurements, four AKUART on the Wall 60 wall absorbers were installed in the dining area prior to Phase 2. Each panel is a 120 × 120 × 6 cm matte white aluminium frame holding a compressed 40 mm glass wool absorber covered in interchangeable and washable polyester-based textile. The panels have an absorption coefficient (aw) of 0.95 according to the ISO 354 standard (ISO, 2009).

The replaceable, sound transparent fabric is customisable for each use case. Keeping in mind the assumption that naturalistic images seem to be preferred among hospital patients [39], we invited the hospital staff to choose four photographs from the AKUART database to print on the canvasses (see Figure 2). The photographs were high-resolution images of a green cabbage, apples, blueberries, and an onion. Staff considered food-related images relevant for the lunch meal context.

After acoustic treatment of the room (see Figure 2), the average speech transmission index was 0.676 alongside an average reverberation time of 0.63 s, much closer to the target value of 0.6 s. The changes in these values, although seemingly negligible, indicate a substantial and noticeable improvement of both the speech intelligibility and the reverberation time of the room (see Table A1).

#### 2.1.2. Music Selection (Phase 3 Only)

Five playlists of instrumental only pieces were compiled. The music was primarily chosen from a list of validated pieces developed in a Danish clinical music therapy project [76,78,79]. The final genres represented in the playlists were Classical, Easy Listening, Folk, Jazz, and MusiCure [80], reflecting a broad range of styles and genres. The sequence of the five playlists to be played on individual days during Phase 3 was decided randomly with no playlists being played on consecutive days (see Figure 2 for audio equipment description). Music was played from the Apple Music application using the built-in feature “Sound Check”, which automatically adjusts playback volume to the same level on all tracks in the Apple Music Lossless audio codec.

#### 2.1.3. Participants

Seventeen patients with acquired brain injury (ABI) indicated by a median score of 99 at the Functional Independence Measure (FIM) scale (11 male, 6 females; mean age M = 64.5 years, SD = 9.2) took part in the study. Of these, 15 agreed to be interviewed about their mealtime experience, and six patients, whose hospital stay spanned two intervention phases, were interviewed twice. Recruitment took place upon admission to the specialised rehabilitation ward.

Criteria for recruitment in the study covered diagnosis of ABI from stroke, traumatic brain injury, anoxia etc. Common changed functional abilities following ABI included cognitive (e.g., memory, attention, and aphasia) and physical (e.g., dysphagia and gross and fine motor skill difficulties). The recruitment criteria for participating in interviews covered the ability to express oneself in Danish, having had at least two meals in the common dining area without severe eating disabilities and assistance needs, and demographic information, such as age and gender, to ensure variation and representability among the participants. 

The patient characteristics are shown in Table 1. All patients meeting the inclusion criteria were informed verbally and in writing about the project upon admission. All patients had the cognitive abilities to give informed consent to participate, as assessed by hospital staff. Consent to participate alongside permission to obtain patient record information was given in written form by the patients. The study followed Danish legislation and the principles of the Declaration of Helsinki and was exempt from ethical approval by the Danish National Committee on Health Research Ethics, due to non-invasive, observational-only data gathering methods.

#### 2.1.4. Meal Procedure

Patients were given a pre-determined meal plan upon admission to the hospital detailing the rotational schedule of the food options. Lunch meals would consist of a soup of the day, a warm meal, followed by open-faced sandwiches with three filling options (such as liver pâté, cold cuts of meat, cheese, etc.). The food rotation scheme was three weeks, after which it would start over. Identical food options in all project phases could thus not be guaranteed throughout the duration of the study. At lunch, patients requested their desired food and drink ad libitum, which was subsequently weighed, registered, and handed out by the staff.

The room was located on the fifth floor of the hospital building, and the room temperature was consistently held at 22 degrees Celsius throughout the project period, which took place during the autumn of 2020.

### 2.2. Data Collection

#### 2.2.1. Overall Sound Pressure Level

Overall sound pressure levels (SPL) during lunch on weekdays in all phases were recorded every minute between 11:30 and 12:15. Two different weightings of these measurements were used: dB(A), which mimics the human-ear frequency response (i.e., the *relative* loudness of perceived sound, primarily in the mid and high frequencies, as the ear is less sensitive to low audio frequencies), and dB(C), which takes into account the entire frequency range, including high and low frequencies, e.g., present in music. Note that decibels are not absolute units of measurements and follow a logarithmic scale.

Acoustic measurements were logged automatically and uploaded online using the 10EaZy 2.8.2 software/hardware system.

#### 2.2.2. Patient Food Intake and Mealtime Behaviour

Weighing of the patients’ food and liquid intake (grams; millilitres); overall observation of behavioural response in relation to the project interventions; the degree and type of social interaction among the patients; body mass index; and FIM were recorded on separate forms developed for the study (see Figure A1). Demographic data, including age and gender, were obtained from patient records.

Behavioural response comprised estimates by the project nurse of (1) the level of social interaction exhibited by the patient, and (2) how the patient responded to the study interventions, if any (e.g., verbally expressed appreciation or criticism about the intervention). Social interaction was defined in this study as: “active, thinking people engaged in meaningful social action with each other”, described by Curle and Keller [81] adapted from Charon [82]. The scale reflected the overall amount of social interaction by each patient in every meal occasion. 

Both social interaction and behavioural influences of intervention were observed by the project nurse and rated on five-point scales; the social interaction scale was anchored by 1 = no interaction and 5 = a lot of interaction. The influence of intervention was judged from negative (1) to positive (5), with the midpoint value of 3 being a neutral “no response” point. The design of both scales was developed in collaboration between staff and researchers. Due to the design of the scales, patients received social interaction ratings during all three phases of the study; in contrast, the influence of intervention was only evaluated in the phases 2 and 3, where some form of intervention took place. The form also allowed for field notes to ensure representation of more detailed accounts of mealtime behaviour and social actions etc. (See Figure A1). For each meal occasion and for each individual patient, values for all measures on the form were recorded.

#### 2.2.3. Patient and Staff Interviews

A series of semi-structured face-to-face interviews with patients were conducted by the first author at the hospital on a weekly basis, depending on patient availability/scheduling. Only the patient and the researcher were present during interviews. No prior relationship between the two existed. Interviews lasted between 10 to 30 min, and each participant signed a separate consent form before the interview. 

The interview guide was developed to assess themes relevant to the study objectives and the participants’ immediate perception/evaluation of the mealtime activity, including how much they enjoyed the lunch meals and the dining room atmosphere; their overall comfort before, during, and after the mealtime experience; their general attitudes towards the project, and their general mealtime and music listening practices (see Table A2). Shorter follow-up interviews with six of the eight patients experiencing the transition between either Phase 1 and 2 (two patients), or Phase 2 and 3 (four patients) were also conducted.

Two interviews with the project nurse were undertaken to supplement patient statements and to provide further observations from the staff point of view. All interviews were audio-only recordings, conducted in Danish, and transcribed verbatim by the first author.

### 2.3. Data Analysis

#### 2.3.1. Sound Pressure Levels

Sound pressure level measurements during the mealtime were analysed using a 10EaZy template for Microsoft Excel. For each meal occasion, two average decibel levels (A-weighting and C-weighting) were calculated based on the 1-min SPL recordings.

#### 2.3.2. Mealtime Observations

The food and fluid intake for each participant were averaged for all meal occasions and grouped by project phase (1–3). Due to non-standard distributions, non-parametric Kruskal–Wallis tests were used to compare mean food and fluid intake values between project phases. In terms of behavioural measures, a Kruskal–Wallis test was run to compare social interaction scores across the three project phases. Finally, one-sample *t*-tests were conducted on response to intervention scores against the no-intervention midpoint scale value of 3, to determine if interventions from phase two or three evoked any behavioural responses from the patients. All quantitative data analysis was carried out with SPSS 28.

#### 2.3.3. Qualitative Interviews

Patient interviews were analysed using a phenomenological/phenomenographical approach to meaning analysis [83,84,85]. Initial automated transcriptions of the raw audio material were made using the NVivo 12 transcription software. The textual transcriptions of the recorded interviews were then edited (e.g., correcting spelling mistakes/wrong words, annotating speakers, etc.). Initial readings formed a naïve understanding of the texts, and overall categories identifying fundamental factors in terms of how patients perceived, experienced, conceptualised, and understood the complex and dynamic mealtime activity were deduced. Associated statements were coded and anchored to these themes for subsequent structuring of the interview topics.

Sections and/or excerpts are reproduced in quotation format translated into English. Quotes are accompanied by an anonymised participant number and the project phase in which the statement was obtained.

## 3. Results

### 3.1. Sound Pressure Levels

The average sound pressure levels (dB) across all phases show that the acoustic treatment reduced the overall sound level in the room from Phase 1 to 2 with −2.02 dB(A) and −2.32 dB(C), and from Phase 1 to 3 by −1.55 dB(A) and −1.12 dB(C). Note that ~1 dB change in SPL results in a noticeable change in loudness perception [45], indicating that the perceived loudness inside the dining room was reduced considerably between the baseline phase and Phase 2. Interestingly, although the overall SPL increased slightly from Phase 2 to 3 with the playback of music, specifically 0.47 DB(A) and 1.20 dB(C), the overall sound pressure level remained below the baseline values, once again confirming the measurable effect of the acoustic panels. Average SPL across all phases are shown in Table 1.

### 3.2. Mealtime Observations

Mealtime observation data are presented in Table 1. A Kruskal–Wallis test with project phase as the independent variable and food and fluid intake as dependent variables was not significant, food: H2 = 0.192, *p* = 0.908; fluid: H2 = 1.918, *p* = 0.383. That said, the average fluid intake increased monotonically in each phase (Table 1).

A Kruskal–Wallis test with project phase as the independent variable and social interaction as dependent variable revealed a significant effect of the interventions. Specifically, the amount of social interaction decreased from the Baseline (Mean Rank = 99.79) to Phase 2 (Mean Rank = 81.85), and subsequently increased in Phase 3 (Mean Rank = 83.62), H2 = 8.745, *p* = 0.013. All patient interactions were, however, reported as being positive in the observation form and no observations of agitated behaviour were made at any point throughout the study.

Finally, a one-sample *t*-test was run to determine whether patients’ response to intervention was different from the no-change score of 3. The mean response score in Phase 2 (4.52 +/− 0.79) was significantly different from 3, with a difference of 1.52 (95% CI, 1.30 to 1.74), t49 = 13.63, *p* ≤ 0.001. The mean response score in Phase 3 (4.9 +/− 0.30) was also significantly different from 3, with a difference of 1.90 (95% CI, 1.81 to 1.99), t49 = 44.33, *p* ≤ 0.001.

### 3.3. Qualitative Interviews

Participants gave nuanced accounts of their mealtime activity at the hospital and were able to identify essential aspects related to the quality of the eating experience. Multiple themes arose from the interviews: the concept of commensality saturated the interviews and was articulated as a significant and indispensable part of the participants’ daily lives (inside and outside the hospital). While difficult to describe concisely, participants conceptualised commensality as a cluster of values including external/physical as well as social components. Another central theme that emerged from patients revolved around the use of music in hospital settings. In particular, patients’ views on appropriate genres for the mealtime context and the affordances that music listening provides are included in the analysis.

In the following, we present these thematic categories as they relate to the project interventions. For the purpose of reporting, these themes will be presented separately; however, a certain degree of overlap exists among the topics.

First, we describe the experienced benefit of the acoustic improvements on the ability to communicate and focus during the meal. Next, we introduce views on the aesthetic component of the sound-proofing panels and the evaluation of the environment following the interventions of Phase 2. Third, we expand upon the central theme of commensality, especially in terms of how the music affected emotional/psychological patient relations during the meal (interaction, conversation). Finally, we evaluate participants’ attitudes and expectations towards the use of music in a hospital setting. Staff perspectives on the above-mentioned themes are included to support the patient statements.

#### 3.3.1. Acoustic Panels Enhance Inter-Patient and Staff Communication

As evident by patient statements, verbal communication was considered an essential part of facilitating the specific communal connection between patients occurring during the common meal. In addition, patients valued calmness and quiet during their lunch, in part to be able to listen and respond to each other while eating, and in part to exclude the commotion of general hospital operations. A patient in the baseline phase described how the bustle of hallway traffic and the busy kitchen area inside the dining room was distracting her from the eating activity: “I think I become disturbed. I am looking around everywhere […] It gives a frustration that it [the kitchen] is over there […]” (Participant 4, Phase 1). 

The same patient experienced the overlap to Phase 2, and, in the second interview, she commented on the comparable differences of the acoustic properties of the room: “It has absolutely helped. It has become more, well more tranquil in the room. The acoustics are better […] there is quieter in the head. We are better able to talk now” (Participant 4, Phase 2). Another patient agreed that the acoustics had improved to a degree that affected the inter-patient communication: “Before, it sounded very… I don’t know how to describe it, was it rumbling in there? Yes, well, it was more ringing. And that is completely gone. When you talk, you can understand everything” (Participant 3, Phase 2). Supporting the patient statements, the project nurse also indicated a noticeable effect of the sound-proofing materials. In particular, she described how being in the room became a more pleasant experience and how certain patients became more actively engaged in the ongoing conversations:

I think there is a big difference in the sound […] I could clearly hear that it was easier to talk at the table…There is less “ringing” in the room […] I think you are much better able to hear what is being said around the whole table. You are taking part more. Because it is not ringing in the same way in there. It is just calmer […] You can almost feel it the moment you enter, even when you don’t speak, it is as if there is just better acoustics. It is just kind of better in a way […] it is a nice place to be (project nurse).

The panels appeared to offer an additional benefit to the staff managing the lunch activity. The project nurse emphasised that communication between nurses and nutritionists also improved during the patient lunch, and that the racket from serving ware diminished:

Before, it was just a bad room in which to move plates and cutlery around, but I don’t think this makes as much noise anymore […] it has also led to us speaking softer. When we’re standing at the buffet, we have not spoken so loudly I don’t think … It has this calming atmosphere (project nurse).

#### 3.3.2. Acoustic Panels Enhance Physical Environment Aesthetics and Promotes “Cosiness” and Pleasantness

A recurring theme throughout the interviews related to being inside a “cosy” space. Participants frequently used the word to positively describe the holistic experience of the mealtime event. For instance, one participant said: “*For me, it’s the cosiness. To eat, that is cosy*” (Participant 06, Phase 1). Cosiness was thought to be ascribable both to inter-personal relations and to the external, physical environment. Regarding the latter, patients emphasised that an attractive dining area was more pleasant and inspiring and had a positive influence on their mood by making the surroundings less institutional. Room descriptions in the baseline phase tended to revolve around the “lifelessness” and “coldness” of the hospital dining environment: “*There is more sterile in there. There’s not that cosiness, that sort of presence that says “welcome”. I think that is missing*” (Participant 05, Phase 1). Another patient stressed this point further:

We are just sitting, you know, like “hospital”, and “older people”, and “it’s a bit sad” and it’s a “serious illness” […] Recovery should be a cosy place where you feel at home, because you have to be there for a long time (Participant 06, Phase 1).

Compared to the baseline phase, patients in Phase 2 appeared to appreciate the appearance of the physical environment to a greater extent. Patients not only noticed the intentional efforts made to decorate the dining area with the photographs, but indeed lauded the homeliness and cosiness of the room: “*I have definitely noticed the pictures that have been put up in there* […] *So they do make it cosy for us* […] *It’s just a cosy place to sit and chat*” (Participant 09, Phase 2), and another commenting: “*It is some very nice pictures* […] *It is important that it is cosy* […] *It means that it is pleasant to be in there. You notice that someone has done something to make it feel pleasant*” (Participant 11, Phase 2). One participant was especially captivated by the motives of the wall panels and illustrated their appreciation by describing a quasi-physical experience that they afforded:

[…] I think that if [the pictures] had not been there, then it would have been very hospital-like. Because there are completely white walls in there. And these pictures are fantastic. The blueberries and the cabbage and the apple. The colours, and then the mere size of them […] It seems cosy, you are almost… it grabs you, the room “hugs” you […] The colours, the deep, red apple and those blueberries and the cabbage, yes, it is a little intoxicating, because of the size. Yes, it is cosy, very cosy (Participant 15, Phase 3).

These statements were echoed by the project nurse who indicated that the perceived advantages of the photographs to make the environment stand out from the rest of the hospital facilities extended beyond the patients’ views. She commented that visitors to the ward, including other hospital staff and management, were enthusiastic about the images and found them extraordinary: “*it has really been much talked-about. Also by people just passing by: “you have put something in there!” These pictures are just attractive, I think*” (project nurse).

#### 3.3.3. Music Enhances the Physical Environment, Prolongs Meal Duration, and the Social Aspects of the Meal Activity

The presence of music in Phase 3 appeared to infuse the room with an aesthetic dimension similar to the effects of the wall panels mentioned earlier. However, whereas the panels provided an obvious visual beautification, the music seemed to offer an improvement to the general mood and atmosphere of the room. The patients considered the music as an integrated part of the dining area that distinguished it from others at the hospital: “*It works well that there is music. It is a room that you feel comfortable inside of. Music makes the room nice*” (Participant 11, Phase 3). 

That music was already playing when they entered the room seemed to have a positive effect as if crossing a threshold into another type of space: “*Well, it sure has been cosy that there was music* […] *It’s not that I think about it before I enter, and then I enter and I see “oh yes, there is music”, and that is very lovely*” (Participant 13, Phase 3). Related, the project nurse recalled a statement from a patient describing her experience of the room after the music had been introduced: “*one of them said one day: ‘it seems like you enter into a little oasis’*” (project nurse).

In general, patients shared the understanding that the music had a calming effect and made them feel immediately at ease. Illustrating this point, one patient stated: “*It’s like this* [shows an embracing gesture with her arms], *like a wave, swaddling you. The music, I mean. Comforting, pleasant*” (Participant 15, Phase 3), while another emphasised that it made him more relaxed: “*You relax more, I think, when there is a little music*” (Participant 14, Phase 3).

Elaborating on this aspect of calmness and pleasantness, the project nurse observed that the music may have influenced the way in which patients physically acted when entering the room, describing: “*from they enter the room, it’s like they almost ‘float’ over to their seat*” (project nurse).

In addition to providing the patients with a relaxing and pleasant atmosphere, there was a general agreement that music assumed a ventilatory role, almost calibrating the shared mood of the room. Music was described as both filling out periods of silence as well as providing a pleasant backdrop to the ongoing conversation: “*I think it has had a positive impact on the people in there* […] *people are sometimes listening to the music. And then we talk a little. And then we listen a little again*” (Participant 13, Phase 3). Closely reflecting this statement, another patient said: “*I enjoy the music when I am in there. I like that it is there. Often there will be pauses you know, and then it’s nice that there is something* [the music] *that takes over, and then the conversation picks up again”* (Participant 14, Phase 3).

There was an additional, somewhat surprising, effect of music playback on the conversational content and quality during lunch. Throughout all phases, having a forum in which to share their lives and everyday experiences was described as very valuable to the patients. Discussing their particular rehabilitation trajectory, in particular obstacles and setbacks along the way, were often the main focus of the talks around the table. Yet, the project nurse pointed out that conversations could tend to circle around the more serious aspects of the hospitalisation experience, sometimes exacerbating a negative mood: “*It can become a bit disease-fixated, even though we try to make it less so*” (project nurse). However, with the introduction of music to the room, the conversational style itself shifted towards other topics. 

This was highlighted by several patients, who described how certain musical pieces evoked positive memories and provided a welcomed distraction from the otherwise serious nature of the hospitalisation: “*You connect* [a piece of music] *with an experience that you’ve had* […] *And that made me think back on how that experience was*” (Participant 14, Phase 3). More patients described how the music itself became the topic of conversations. Specifically, patients actively engaged in discussions about the music, either commenting on the pieces or guessing the names of the songs or artists as they were played:

I definitely think there are more conversations now. We talk about ‘this one, we know this one’, and ‘this one is by so-and-so’ […] And I think that the chatter is going better […] there was a melody from a gymnastics show I had seen, that I really liked. Then I told the others about that experience I had had with that piece of music. In that way the music leads us onto some other topics than we normally talk about (Participant 14, Phase 3).

Another patient repeated the notion that “[the music] *is also a conversational topic for us: ‘Is it good music today?* […] *Oh, it’s waltz music*’ […] *the music, now that I think about it, is something that we comment on every day*” (Participant 15, Phase 3). One patient expressed that the mealtime music to her was a way of bonding with people with whom she would otherwise not share many commonalities:

[…] it is nice that there is background music […] we did not choose each other. In that way, it is nice that there is also music. Then you have something to naturally talk about […] it has definitely generated conversations about the music (Participant 11, Phase 3).

Finally, a number of statements by the patients who experienced the overlap between Phases 2 and 3 indicated that the pleasant and relaxing mealtime atmosphere created by the music made them want to stay in the room beyond the normal lunch duration. They noted that they stayed up to 15 min past the time when lunch would normally conclude: “*There were days when we were in there an hour, and we have not done that before as far as I remember. So, it has been longer days, maybe 10–15 min more some days than before*” (Participant 13, Phase 3). This notion recurred in another patient statement: “*Before the music, we just ate and then we left. Now we stay almost until 12:15 every day. Otherwise, we would leave around 12, so yeah in that way, we stay a little longer I think*” (Participant 14, Phase 3). Patients also shared the view that the music somehow affected their eating behaviour. Specifically, they mentioned spending more time eating when the music was playing, as well as experienced increased enjoyment from the food. One patient stated that:

I think it [the music] does that you take longer to eat. Because you are sitting and eating and enjoying […] The way that we eat the food. It is better digested than when we just quickly go in and eat and go back to our rooms. So, we sit and listen to the music and then enjoy (Participant 12, Phase 3).

This view was brought forth in another interview: “*that thing when you sit and eat, they you take a small break to just listen. And then that thing where you just take a cup of coffee and such …*” (Participant 15, Phase 3). A similar observation was made by the project nurse, who stated that the additional minutes spent inside the room also led people to reach for supplemental food items:

They have been sitting in there much longer. That has really been thought-provoking […] It was this thing where they would have a cup of coffee and talk a little. It was much more a cosy atmosphere after the meal […] and some of them actually eat more. It’s more this thing of where they grab a piece of fruit or a piece of bread with the coffee (project nurse).

#### 3.3.4. Patient and Staff Views on Music in Hospital Settings

Throughout the interview process, most patients brought to light the ubiquitous presence of music in their everyday lives, often described as underscoring a host of activities, affording specific moods, or used to modulate emotional states. Participants also pointed out that they regularly used music as an accompaniment to eating and in general, they overwhelmingly linked their use of self-chosen music to mood-improvement and relaxation: “*Well, it makes me happy* […] *music to me, that’s joy*” (Participant 03, Phase 1), “*It makes me relax and be happy*” (Participant 04, Phase 1), “*You become happy*” (Participant 08, Phase 2), “*It makes you happy. You are never angry when you listen to music*” (Participant 10, Phase 2), “*It gives a sort of optimism. And I think you feel good. It is a joy of life*” (Participant 12, Phase 2). 

Despite their everyday habitual music listening, patients did not expect to encounter music in the hospital setting, although there was a general belief among the patients that music would improve the mealtime experience: “*That* [music at the hospital], *I would say, I would not expect* […] *But I think it could be a nice thing*” (Participant 12, Phase 2), “*I don’t have a certain expectation about it, but I think it would be nice if there was like a quiet background* [music]” (Participant 05, Phase 1).

Concerns about the prospect of music before Phase 3 were expressed by a few patients. They were mainly anticipating that the loudness of the music might interfere with the ability to talk at the table: “*I think it would be a pity if the music became so loud that we almost couldn’t speak*” (Participant 13, Phase 2). However, these same participants responded positively to the transition from Phase 2 to 3, explaining: “*I think it has been at a suitable level. I was very worried that it would be too loud. And I don’t think it has been* […] *it worked well, and it has been positive*” (Participant 13, Phase 3). 

In fact, they even indicated that the music occasionally had been indistinct: “*There has been a couple of days where it was too soft. I would have thought it would rather have been too loud, but there were days where it was too soft*” (Participant 11, Phase 3). The project nurse confirmed that patients had briefly commented on the music playback volume, while at the same time noting that it did not seem to influence their ability to focus on the meal. In fact, she stressed that the overall response to the music had been overwhelmingly positive, and that the volume was ideal for both conversation and music listening.

In terms of the music genres represented throughout Phase 3, the majority of participants emphasised variety as a positive element. Surprisingly few patients had negative opinions on the music, and there was an overall acceptance of the styles. Patients identified the playlists they enjoyed the most and were able to describe how they paid scant attention to the pieces they enjoyed less: “*It’s like I kind of ‘cut’ out the music and say ‘well, this is not the important part*’ […] *You can shut off something you don’t think is good*” (Participant 14, Phase 3). 

Only in one patient interview, the jazz playlist was specifically associated with negative outcomes, whereas the classical playlist was perceived as very comforting: “*It* [jazz] *gives disquiet in the brain. It interrupts. And then it removes focus from why you are in there. It was better with something like Mozart, something like that. I mean something that swaddles nice and soft*” (Participant 15, Phase 3). The project nurse made a similar observation, noting that certain patients appeared to be more sensitive to music with a higher dynamic range and/or conveying apparent emotional content. She recalled patients commenting on the “gloominess” of particular classical pieces or the “sleepiness” of certain MusiCure pieces, in addition to observing the somewhat distracting nature of the jazz playlist. She explained that the music “*should not be either too jolly or too gloomy* […] *It needs to be ‘neutral’ in the mood*” (project nurse).

From the five playlists, the one comprised of well-known songs in instrumental versions was often mentioned as the most preferred playlist, in particular because it was the one capable of creating associations to previous positive memories and gave rise to conversations. As previously mentioned in Section 3.3.3, familiar pieces gave the group a shared point of departure for talking about their experiences. One of the patients commented on the playlist, saying: “*it was actually ones that we all knew* […] *Stuff like that you can just sit and enjoy really much, especially if it’s a good piece that you liked once*” (Participant 14, Phase 3).

## 4. Discussion

Suggestions that music can improve various health-related outcomes in the healthcare sector are not new, and the empirical evidence documenting the beneficial effects of music on physical and psychological disorders is rapidly mounting [51,62,86]. However, research on the implications of using music and/or the significance of the acoustic environment on the hospital mealtime experience is scarce. This study explored the effects of aesthetic interventions, i.e., acoustics and music, to a hospital mealtime environment at a specialised ward for patients with acquired brain injury. Several perspectives were addressed combining the assessment of data collected and analysed using various methods both quantitative and qualitative.

### 4.1. Acoustic Panels Enhance Inter-Patient and Staff Communication

The first objective of the study was to identify ways of optimising the acoustic properties of the dining area. The acoustic treatment decreased the overall sound pressure level during lunch, decreased the reverberation time, and improved the intelligibility of speech from the baseline to Phase 2. Even with music playback in Phase 3, the overall sound pressure level was kept below the levels obtained in the baseline phase. These results indicate that acoustic improvements to existing hospital architecture are indeed both possible and highly effective and should be considered on a wider scale to decrease the risk of noise-related health-issues as reported by numerous studies [47,53,54,55,56,57].

### 4.2. Nutritional Observations

The second objective was to assess the impact of the acoustic room treatments, visual panels, and music during lunch on the nutritional status of patients in terms of intake amount, as registered by the caring staff. We found no significant effects on overall consumption of food or fluid, while we did observe a non-significant monotonic rise in beverage intake from the baseline phase to Phase 2, and from Phase 2 to 3.

Of interest is the subjective accounts describing additional food or drink items being consumed in Phase 3 when music was playing. These anecdotal accounts are in line with previous studies demonstrating that dinner music is associated with longer mealtimes and higher food intake among dementia patients [7,8]. Therefore, future studies should investigate whether mealtime music has potential long-term effects on overall food intake on larger sample sizes and among patient groups that may experience difficulty eating, loss of appetite, or disordered relationships with eating.

### 4.3. Acoustic Panels Enhance Physical Environment Aesthetics and Promotes “Cosiness” and Pleasantness

Based on the findings by Beck and her colleagues, mealtimes can be considered as an opportunity for the patients to experience well-being when in a calming and appealing environment [9]. We found that the wall panels improved the physical attractiveness of the dining area according to the patient accounts. Patients experiencing the overlap between Phase 1 and 2 appreciated the homeliness, beauty, and comfort the wall panels provided, reducing the “sterility” of the environment and acted as positive distractions from the institutional nature of the hospital environment. 

Our results appear to be in agreement with this research suggesting the importance of “aesthetics” in mealtime settings and supports existing evidence arguing that the presence of visual art in hospitals can help reduce symptoms of stress and improve mood during hospitalisation [10]. Research previously found that naturalistic images tend to be favoured by mental health patients, because abstract artworks may offer unnecessary cognitive load to the patients [11]. Measuring the impact of the panels on psychological or nutritional aspects alone was outside the scope of the present research. However, the appreciation of the photographs by the patients were substantial and supports the notion that visual art can contribute positively to patients’ mood and feelings of comfort. Therefore, we recommend further research investigating whether such perceptual impacts could translate to other health aspects related to food and eating.

### 4.4. Music Enhances the Physical Environment, Prolongs Meal Duration and the Social Aspects of the Meal Activity

In terms of the behavioural measures obtained from the registration forms, we found that patients responded significantly more positively to the music intervention than the acoustic intervention, compared to the baseline reference. Moreover, data from the registration forms revealed that the amount of social interaction decreased from the baseline phase to Phase 2, before increasing again in Phase 3. In addition, the qualitative interviews highlighted a shared understanding that the music provided calmness, relaxation, and positive mood to the group during lunch. We argue that this extends the previously mentioned notion that aesthetic properties in the surroundings promote a sense of homeliness and familiarity and affords a notion of comfort and safety during the meals [9,12,13] to include music as one such aesthetic element.

In addition to providing physical calmness and enhancing the mood during lunch, patients reported that the nature of the social interactions during lunch also appeared to change in the third phase. While the presence of music may not have increased the amount of patient interaction, it contributed to more positive relations during the meals and provided new content for conversations. Our results correspond to previous research demonstrating that mealtimes are important social activities for patients, and a place where a sense of community and cohesion can be developed and maintained throughout the hospitalisation period [9]. 

However, previous studies have indicated that hospital mealtimes tend to be characterised by the influence of disease, and that patients’ willingness to eat can be negatively impacted by a focus on illness during the meal [14]. Our observations of the improved mood during the lunch meal in Phase 3 point towards music as another factor to potentially distract from pain or disease-related topics of conversation and should be considered a staple of the hospital eating environment on a broad scale. From a nutritional perspective, earlier research findings suggest that social environments conducive to interaction, in particular meal fellowship, is positively associated with a higher amount of food eaten [15]. 

Possible explanations for the mechanism underlying this relationship are often ascribed to the theory of social facilitation [16], in which subjects consume more in the presence of others, typically as a result of prolonged meal duration due to a more sociable atmosphere [15]. However, it has also been argued that it is not only the prolonged meal duration but also the particular nature of the individual behaviours that drive this effect, specifically positive interactions [17]. Based on the findings of our study, we argue that music not only contributed to longer mealtime duration but also facilitated social interaction of a more positive character during the meals.

### 4.5. Patient and Staff Views on Music in Hospital Settings

The findings from the interviews indicated that patients above all enjoyed the presence of music and the variety of the genres, and that individual music preference carried secondary importance in the overall experience of the music during lunch. This points towards the possibility of music to enhance the mealtime experience regardless of personal taste. We did find evidence of less preferred musical styles, specifically the jazz playlist and a few of the classical pieces, which seemed to be too energetic, emotional, and/or dynamically fluctuating for the mealtime situation. These results correspond to findings from existing literature arguing that mealtime music should encompass the characteristics of the existing definition of “relaxation music” found to decrease adverse behaviour of dementia patients [63,73,74,75,79]. 

However, contrary to the existing belief that familiar music could have negative impacts on patients [67] and that mealtime music should be unrecognisable to reduce the cognitive load of trying to identify the pieces [65], our findings show that familiar music (e.g., instrumental versions of known songs) provided patients with conversational topics and contributed to better social bonding and interaction in addition to evoking pleasant memories. It should be pointed out that the patients in our study did not suffer from severe cognitive impairments or dementia and were, thus, less likely to react negatively to more stimulating music. This exposes the need for future studies to take into account the specific characteristics of the patient group when selecting hospital mealtime music playlists.

### 4.6. Limitations

To our knowledge, this is the first study to comprehensively investigate the effects of specific environmental interventions on hospital mealtime experiences and nutritional outcomes, incorporating both quantitative and qualitative data acquisition methods. However, there are obvious methodological limitations to the study design and uncertainties tied to the findings.

An absolute between-subjects design could not be achieved due to the overlap of patients between phases. In addition, it should be noted that our ability to recruit patients was impacted by Covid-19 restrictions in terms of the dining room capacity. Only four patients were allowed in the room for each meal occasion, resulting in a small sample size and potential limited generalisability of the findings. However, due to a relatively homogenous study sample in terms of diagnoses and physical/cognitive characteristics, the results may be transferred to similar contexts. 

We must also point out that certain subjective accounts may be examples of after-the-fact intellectual speculation, rather than accounts of the actual situated effects of the interventions. All interview data were obtained and analysed by the same researcher, which could introduce the possibility of misinterpretations in the process of analysis. That said, considering the high degree of consistency across patient testimonies, this is unlikely to have played a role in the interview analysis process.

Likewise, mealtime observations and staff reflections were made by the same project nurse, which increases the risk of evaluation bias. Overestimation of behavioural response and social interaction scores can, therefore, not be ruled out. However, knowing the patients’ traits and behaviours in the dining situation enabled the nurse to observe more detailed comparisons of patients’ experiences across phases, in addition to being known and trusted among the patients.

Regarding food-related measurements, it could be argued that our measures of nutritional status and food intake alongside the subjective accounts could be further qualified by including more precise, yet non-invasive instrumental measures, for instance bio impedance analysis or other biochemical markers of nutrition. In addition, heed should be taken to the day-to-day variation in meals throughout the project period making comparative evaluations of the meals impossible. 

Furthermore, our study did not investigate the potential impact of the music on the tasting experience of the food itself. Evidence has long been indicating that background sound can influence the way we eat and how the food tastes [87]. For example, specially designed soundtracks can modify taste evaluation [88] and eating speed [89,90,91,92]. While this has predominantly been investigated in healthy, young populations, there are reasons to believe that effects, such as these could be found in hospital settings as well. Further research is warranted to examine the potential effects of acoustic interventions and music on food enjoyment and taste perception in a hospital meal context.

Future studies should first and foremost prioritise larger sample sizes and focus on employing more precise, systematic, and rigorous data collection methods to consolidate our results and expand the applicability of our findings.

### 4.7. Implications

Our study demonstrates that a quality improvement approach of simple infrastructural modifications to the dining area can provide an important foundation for improving the mealtime experience for patients. This has obvious practical implications for the healthcare sector.

First, the positive findings regarding the installation of acoustic sound absorption panels documented in this study provides hospital management with an incentive to consider installation of noise dampening materials as a cost-effective strategy to reduce potential negative outcomes related to noisy hospital eating environments.

Secondly, increasing meal duration by simply playing music may provide hospital staff with an opportunity to encourage greater food intake during common lunch meals, especially among patients at risk of mal- or undernutrition. In addition, mealtime music should not be overlooked as a safe, simple, and efficient method to encourage social interaction, enhance mood and well-being during eating, and distract from stressors, which could prove helpful in encouraging more expedient food intake among those struggling with eating, including dysphagia or cancer patients, or those with disordered relationships with food, such as anorexia patients.

Furthermore, the study provided the opportunity to engage patients and staff in the process of shaping the future hospital dining experience as well as identifying appropriate selections of music for a mealtime setting by asking participants about their everyday music listening practices as well as their expectations around the use of music in a hospital context. We propose that a broader range of stylistic variations than “relaxation music” alone, and even music of a more stimulating nature, may be of particular benefit to the hospital mealtime experience. It may well be that, for some patient groups, the mechanism by which music acts during meals is not solely due to the inherent musical characteristics but can also be attributed to situational factors, such as distraction from the hospitalisation experience, as previously mentioned.

Finally, the project described here demonstrates the benefits of inclusive, collaborative efforts between stakeholders from state and local organisations as well as private industry professionals, opening up avenues for future research endeavours to improve the healthcare system.

## 5. Conclusions

This exploratory project demonstrated that data-triangulation using instrumental measurements, observational quantitative, and subjective qualitative data offered nuanced insights into the relationship between the dining environment and the mealtime experience of patients during in-hospital rehabilitation following ABI. Ameliorating acoustic room properties and playing music during meals improved ability to converse, facilitated better social bonding and conversations, extended the duration of the meal, and exerted a calming influence on patients and staff.

First, the positive findings documented in this study provide hospital management with an incentive to consider installation of noise dampening materials as a cost-effective strategy to reduce potential negative outcomes related to noisy hospital eating environments. Secondly, increasing meal duration by simply playing music may provide hospital staff with an opportunity to encourage greater food intake during common lunch meals, especially among patients at risk of mal- or undernutrition. In addition, mealtime music should not be overlooked as a safe, simple, and efficient method to encourage social interaction and enhance mood and well-being during eating.

Taken together, the project provides a practical approach to obtain optimal acoustically conditioned hospital eating environments using attractive innovative means, i.e., acoustic wall absorption panels and mealtime music, thereby, increasing the overall quality of the patient mealtime experience.

## Figures and Tables

**Figure 1 foods-10-02590-f001:**
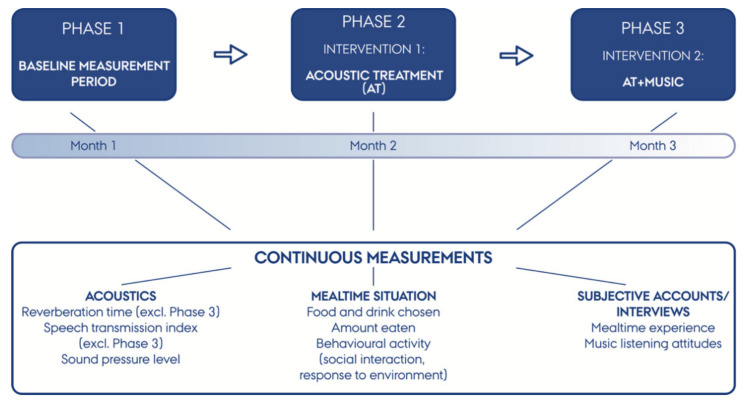
Experimental study design: project duration, intervention phases, and data measurement strategies.

**Figure 2 foods-10-02590-f002:**
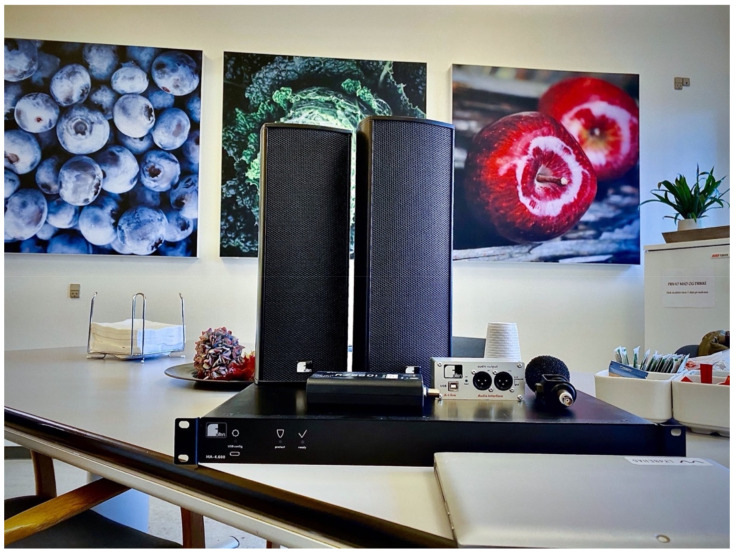
Acoustic and audio equipment. Panels: AKUART On the Wall 60 wall absorbers. Audio equipment: Fohhn Audio Scale-1 speakers; Fohhn Audio MA-4.100 amplifier; Fohhn Audio A1-Live USB-sound card, and MacBook Pro laptop.

**Table 1 foods-10-02590-t001:** Characteristics of study participants, mealtime observations, and the average sound pressure levels.

Characteristics of Study Participants	Total	Phase 1 Baseline No Intervention	Phase 2 Acoustic Intervention	Phase 3 Acoustic Intervention + Music
Number of patients (overlapped from previous phase)	17	7	11 (4)	7 (4)
Male	11	5	8 (3)	4 (3)
Female	6	2	3 (1)	3 (1)
Mean age in years (SD)	64.47 (9.19)	67.29 (7.65)	65.55 (8.29)	60.50 (11.20)
Mean BMI (SD)	28.33 (5.01)	27.14 (3.02)	30.13 (5.47)	26.87 (8.03)
Mean FIM * (SD)	99.35 (20.55)	97.14 (21.82)	96.43 (22.57)	111.33 (12.66)
Mean no. of meals in the dining room (SD)	10.35 (5.89)	8.14 (5.64)	5.73 (2.24)	7.57 (3.69)
Mealtime observations				
Average food intake in grams (SD)		334.96 (107.87)	359.70 (99.83)	338.50 (100.31)
Average fluid intake in millilitres (SD)		282.42 (60.89)	327.61 (143.42)	342.23 (89.32)
Average response to interventions score (SD)		3	4.52 (0.79)	4.9 (0.30)
Average social interaction score (SD)		4.77 (0.95)	4.30 (1.49)	4.60 (0.84)
Average sound pressure levels (SPL)				
Average sound pressure levels, dB(A)		64.49	62.47	62.90
Average sound pressure levels, dB(C)		67.85	65.53	66.70
Difference from baseline, dB(A)			−2.02	−1.55
Difference from baseline, dB(C)			−2.32	−1.12

* FIM = Functional Independence Measure-scale: 18-item ordinal scale used to determine need of assistance of patients with diagnoses within rehabilitation populations. Scores range from 1 to 7. Scores are added for each item with total possible scores ranging from 18 (lowest possible) to 126 (highest possible) level of independence.

## Data Availability

The data presented in this study are available on request from the corresponding author. The data are not publicly available due to personal identifiable information obtained via patient records.

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
