# Peer review of "Music Is Served: How Acoustic Interventions in Hospital Dining Environments Can Improve Patient Mealtime Wellbeing"

_foods, 2021, doi:10.3390/foods10112590_

Round 1
Reviewer 1 Report
The manuscript has been submitted for publication to Foods by Mathiesen et.al, titled "Music is served: How acoustic interventions in hospital dining environments can improve patient mealtime wellbeing" is aiming to measure the impacts of acoustics and music on the lunch mealtime in acquired brain injury (ABI) patients.
The manuscript discussed an interesting phenomenon but produced limited novelty and scientific.
Some points that in the reviewer's opinion were the weaknesses of the manuscript as below:
1. From an epidemiological standpoint, the research design was described as similar to a quasi-experimental study. Moreover, it did not have any control group. However, the authors did not present any effort to figure out and control the potential confounders. And the sample size was small.
2. The methods in no way warrantied that the outcomes changes were mainly due to the impacts of the interventions for acoustics and music. For example, the improvements in mealtime and social interaction might be the consequences of ABI treatments and healthcare.
3. The dining room environment factors relevant to patient well-being should be considered, such as the differences in temperature by season or air-condition, or light intensity.
4. Measurement of patient behavior and social interaction was conducted by the project nurse, it could lead to the overestimation of results a kind of information bias.
5. Followed the pre-post measurements, the statistical strategies with a Kruskal-Wallis test and a Mann-Whitney test was not appropriate for analyzing paired-data
6. There was an ethical concern in the participants recruited procedure. Did all ABI patients have fully cognitive to respond to the consent form and qualitative interviews?
7. There were some additional issues, such as Was the study approved by IRB? Were the measurements tools validated? Was any content validity test, sensitivity test conducted?
Reviewer 2 Report
Content:
This is a well-written manuscript reporting a mixed-methods study in 17 patients with brain injury. The interventions were acoustical measurements and the application of music. The authors obtained data on sound pressure levels, mealtime observations and mealtime behavior, and perceptions of patients regarding the dining room and the relation to other patients.
Major Comments:
The introduction explains that studies have been performed to test the effect of music in recovering patients, but the acoustic environment has been neglected as an important influencing factor. The methods section is detailed and provides an insight into all the measures that have been taken and obtained to generate data. Table 1 summarizes data on the patient group, mealtime observations and the acoustic surrounding field. Additional tables and figures in the appendix help to understand the methodological approach. The results section is clear and comprehensible, and the discussion integrates the findings into the literature. It is interesting that the intervention did not only change the eating behavior and the length of meals but also the social interaction of people during meals. The limitations of the study are discussed.
Minor suggestions:
50% of the abstract is information that could be obtained from the literature. However, important aspects of the study design such as the number of included patients and the diagnosis of patients are missing. Thus, the abstract needs to be re-written to include more study-specific information.
The discussion is very much focused on the results of the study. However, the results might have relevance for future studies in other brain-related disorders or disorders related to eating, for example in anorexia nervosa. The authors could very briefly mention studies on the attitudes surrounding music of patients with anorexia nervosa or studies on the effects of music in people with anorexia nervosa and explain how adding the additional factors they had in their study (controlling or influencing the acoustic environment and measuring the effect of music on social interaction) might lead to improvements in these areas of medicine. Such a short paragraph would show the bigger picture regarding the relevance their approach might have in medicine where eating is a problem such as cancer cachexia, anorexia nervosa etc.
Reviewer 3 Report
I'd like to thank the authors for the opportunity to see their paper. I really enjoyed the reading and I found the results very interesting. I think the methods are adequate for the study and the results are well reported. My only concern is about the description of the quality data analysis that should be implemented. I really appreciated the 3 steps approach applied.
Overall, I think the paper is suitable for the journal and deserves to be published.
Round 2
Reviewer 1 Report
The authors have reasonably addressed the reviewer's comments/points.